# Anomaly Detection in Biological Early Warning Systems Using Unsupervised Machine Learning

**DOI:** 10.3390/s23052687

**Published:** 2023-03-01

**Authors:** Aleksandr N. Grekov, Aleksey A. Kabanov, Elena V. Vyshkvarkova, Valeriy V. Trusevich

**Affiliations:** 1Institute of Natural and Technical Systems, 299011 Sevastopol, Russia; 2Department of Informatics and Control in Technical Systems, Sevastopol State University, 299053 Sevastopol, Russia

**Keywords:** anomaly detection, machine learning, biological early warning systems, mussels

## Abstract

The use of bivalve mollusks as bioindicators in automated monitoring systems can provide real-time detection of emergency situations associated with the pollution of aquatic environments. The behavioral reactions of *Unio pictorum* (Linnaeus, 1758) were employed in the development of a comprehensive automated monitoring system for aquatic environments by the authors. The study used experimental data obtained by an automated system from the Chernaya River in the Sevastopol region of the Crimean Peninsula. Four traditional unsupervised machine learning techniques were implemented to detect emergency signals in the activity of bivalves: elliptic envelope, isolation forest (iForest), one-class support vector machine (SVM), and local outlier factor (LOF). The results showed that the use of the elliptic envelope, iForest, and LOF methods with proper hyperparameter tuning can detect anomalies in mollusk activity data without false alarms, with an F1 score of 1. A comparison of anomaly detection times revealed that the iForest method is the most efficient. These findings demonstrate the potential of using bivalve mollusks as bioindicators in automated monitoring systems for the early detection of pollution in aquatic environments.

## 1. Introduction

An objective assessment of the state of aquatic ecosystems is impossible without the use of biological methods of environmental monitoring. Timely detection of the possibility of emergency situations by biomarkers [1], in the vast majority of cases, allows for the implementation of measures to prevent damage to the environment and reduce the consequences of their impacts [2]. This is especially important for the water supply systems of cities and large settlements. It is equally important to carry out such control in the areas of exhaust manifolds of cities and industrial enterprises. The existing control systems, based mainly on physical and chemical methods, are laborious, expensive, provide fragmentary information, cover the traditional narrow range of pollutants, and do not provide continuous monitoring and timely detection of a sudden release of pollution. In 2008, an automated biomonitoring system was developed by the authors, which is an analog of the Musselmonitor^®^ system, designed for operation in natural conditions of reservoirs [3]. The work of the system is based on fixing and analyzing the behavioral reactions of mollusks and generating an alarm signal when an anomaly is detected. Devices operating on this principle are called biological early warning systems [4,5,6,7,8,9]. In bivalve mollusks, the magnitude of the opening of the valves and the features of the rhythm of their movements characterize the filtration activity and, consequently, the level of their vital activity in normal and toxic environments [10].

Anomalies in the behavioral reactions of mollusks can be identified by different methods, one of them being machine learning algorithms [11]. Machine learning methods allow tuning (training) algorithms using some training data set to solve various problems. Machine learning algorithms have been increasingly used for classification and clustering in the assessment of environmental parameters by biological systems in recent years [12]. Examples of using machine learning algorithms for behavior detection in shellfish activity and anomaly detection are given in Section 2.

The range of possibilities for using machine learning for anomaly detection in the activity of mollusk valves has not been studied enough. Hence, the objective of this paper is to investigate the feasibility of using four traditional unsupervised machine learning algorithms for anomaly detection in the behavioral reactions of mollusks in automated biomonitoring systems of aquatic environments. The anomaly detection technique was carried out for its subsequent inclusion in the software of existing and real-time automated biomonitoring systems of aquatic environments.

## 2. Related Work

Various machine learning algorithms are applied to the study of bivalve mollusks in solving certain problems; for example, to detect mussels contaminated with heavy metals according to spectroscopy data [13]. Algal blooms have a negative impact on aquaculture and drinking water supply and are an environmental problem. The study [14] showed the effectiveness of using machine learning algorithms to identify algal bloom drivers, which are the source of toxin that accumulates in shellfish tissues. For this purpose, the authors used the random forest method to classify shellfish above and below a threshold of toxicity [14]. Machine learning techniques have been used to generate a cyanobacterial bloom alarm by detecting anomalies in phycocyanin fluorescence data without the need for an appropriate cell count or biovolume [15]. The review article [16] shows a variety of machine learning techniques being applied to develop effective tools to help shellfish farmers manage and anticipate harmful algal blooms and shellfish pollution events, which often result in significant negative economic impacts. Six machine learning algorithms were applied to build a predictive model for the closure/opening of the production areas of cultivated mussels in Galicia (Spain) when a critical concentration of marine bio-toxin [17] is detected due to active algal blooms. The kNN method showed the best result, and the developed models, according to the authors [17], can be used to assess the reliability of decisions made by experts. Using remote sensing data, Hill et al. [18] showed the feasibility of using machine learning models to detect and predict harmful algal bloom events in the Gulf of Mexico. Several machine learning algorithms (multiple autoregressive and artificial neural network (ANN) models) have been successfully applied to predict shellfish contamination with diarrheal shellfish poisoning (DSP) toxins in shellfish production areas in Portugal (Cruz 2022 [19]). Wang et al. [20] conducted a classification of eleven types of algae that produce paralytic shellfish poison. Grasso et al. [21] showed the ability of predicting biotoxin contamination in shellfish by predicting PSP toxin concentrations in blue mussels using a deep learning algorithm (e.g., a single hidden layer FFNN model).

An unsupervised method (random forest) was used to find patterns in relative concentration data of polycyclic aromatic hydrocarbons in mussels of the *dreissenid* family [22]. The SVM algorithm was used for classification to estimate the boundary of the ecological niche of zebra mussels (*Dreissena polymorpha*) in North America [23]. The random forest algorithm was used to test if the genetic differentiation of Mytilus mussel populations may be related to any of the key environmental variables known to shape species distributions [24]. Valetta et al. [25] showed the possibility of using machine learning algorithms in animal behavior studies, and later Bertolini et al. [26] successfully applied unsupervised machine learning algorithms (k-means clustering) to identify consistent behavioral patterns in the activity of bivalve mollusks *Mytilus galloprovincialis* and *Mytilus edulis*. Two machine-learning algorithms (support vector machines and classification trees) were used to assess the group classification accuracy of two phenotypes of *Lampsilis* teres (Keogh [27]).

Machine learning algorithms (both traditional machine learning and deep learning approaches) are widely used for anomaly detection and prediction of water quality (drinking, aquaculture, natural water bodies) in real time [28,29,30,31] based on data from sensors of physical and chemical indicators (temperature, pH, etc.). Based on three water quality parameters (such as ammonia nitrogen, turbidity, and electroconductibility) and using the developed IGA-BPNN model in the case study on the Ashi River of Songhua River Basin, China, the authors of [29] showed that the model can effectively reflect the isolated sharp peaks of the water quality parameters and guarantee the efficiency of early warning. In [31], six machine learning methods (SVM, RNN, DNN, and others) are tested to find the best model for anomaly detection on water quality systems based on water quality sensors in Germany. The authors conclude that all methods are vulnerable. Shi et al. [32] proposed a combined approach of a wavelet artificial neural network (wavelet-ANN) model and high-frequency measurements from sensors to anomaly detection for surface water quality management on the monitoring program applied to the Potomac River Basin in Virginia, USA. Later, Liu et al. [33] used the isolation forest algorithm for surface water quality anomaly detection for the early warning of the large-scale release of potentially harmful substances resulting from spills into the river or intentional releases. Machine learning algorithms (deep neural network DNN) are also being applied to predict the abundance of *Dreissenid* mussels in coastal waters using underwater images [34]. Machine learning and classification algorithms made it possible to process and extract informative expression signatures from high-dimensional mass-spectrometry data using the example of a case study of oil pollution in the mussel (*Mytilus edulis*) [35].

Most of the works described above detected anomalies using machine learning methods based on data from physical and chemical sensors for monitoring the quality of the aquatic environment. The difference in this work lies in the use of experimental raw mollusk reaction data (valve opening value in millimeters) for anomaly detection and alarm isolation.

## 3. Materials and Methods

Data for the anomaly detection of the water conditions of the Chernaya River (Sevastopol region, Crimean Peninsula) (Figure 1) were obtained using the system for automated monitoring of the aquatic environment developed by the authors based on the behavioral reactions of bivalve mollusks [3,36]. Freshwater mollusks *Unio pictorum* (Linnaeus, 1758) were used as bioindicators. Freshwater mollusks are attached with one valve to the site of the underwater part of the device with a polymeric adhesive composition, and the second valve of the mollusk acts on a flexible plate with a permanent magnet attached to it, capable of moving freely. The Hall sensor is located under the attachment area of the mussels. Changing the distance between the valves during their movement and, accordingly, the distance between the Hall sensor and the magnet, changes the output voltage of the sensor. This voltage after amplification is digitized and transmitted via a GSM channel to a dedicated server. Thus, the data are the values of the mollusks valve opening value in millimeters with a period of 10 s (an example of raw data is presented in Appendix A).

The developed system made it possible to install up to 16 mollusks, but during operation, 2 of them failed (numbers 1 and 16). Therefore, the data on the activity of the valves of 14 mollusks were used to train the models.

When using machine learning methods, it is recommended to evaluate several algorithms and compare their performance to select the best model that solves the problem [37,38]. Four unsupervised machine learning algorithms for anomaly detection were applied to the observed data on the behavioral responses of mollusks: local outlier factor (LOF), one-class support vector machine (SVM), elliptic envelope and isolation forest (iForest).

The period from 26 February to 24 April 2017 was selected for anomaly detection in the data series of bivalve mollusk activity, during which a violation of the monotony in the daily cycle of mollusk behavior was found during the period of intense rains in the catchment area of the Chernaya River. The exact time of occurrence of the anomaly is unknown. However, the method of expert evaluation revealed the days on which anomalies occurred. At this moment, there was a sharp increase in the frequency of unsystematic short-term clamping (up to 2 per minute) and a decrease in the amplitude of the opening of mollusk valves for 2–3 days (Figure 2). In Figure 2, different colors (abbreviations M2, M3… M15 correspond to the number of the mollusk) show the daily activity cycle of the bivalve mollusks used in the system. At the same time, in 20–25% of mollusks, complete closure of the valves is noted for from several hours to a day. During this period, a decrease in water temperature by 2–4 degrees and an increase in water turbidity were recorded, according to the data of the laboratory for water quality monitoring. It is quite likely that toxicants from adjacent agricultural fields, where fertilizers and pest control agents were used, entered the riverbed along with soil washouts, which explains such an intense reaction of mollusks.

We split our data set into subsets with a 5-day time interval training set, and in each iteration, a 1-day subset was used for the test. The choice of a 5-day training interval was a compromise, as increasing the interval would have increased the time between two potentially distinguishable possible anomalies, and decreasing the interval would have resulted in a decrease in training data. After the discovery of the anomaly, we did not consider the next 4 days. Overall, we had 38 subsets. Each 5-day training subset was a 43,200-point time series for each of the 14 mussels. Each 1-day test subset was a time series of 8640 points for each of the 14 mussels. The amount of training and test data depended on the averaging value, for example, with 30-min averaging, the amount of test data was 48, and without averaging—8640. The data averaging time was used as one of the hyperparameters of the models. Each model used its own algorithm to detect anomalies in the data, so the optimal averaging value could only be obtained experimentally. We used the following averaging values: no averaging, 1 min, 5 min, 15 min, and 30 min.

The general scheme of the action sequences of the models is presented in Figure 3. After splitting the data into training and test samples, the procedures for scaling and averaging the data were carried out. Then, a hyperparameter tuning process was carried out for each machine learning algorithm in order to maximize the F1 score and minimize type II errors. 

Anomalies in the mode of behavioral reactions of mollusks are distinguished due to natural processes, for example, heavy rains, and due to technical malfunctions of the automated system. The work is aimed at anomaly detection of both types.

The efficiency of machine learning algorithms was evaluated by the F1 score [39], an integral indicator that is the harmonic mean of the recall and precision of detection:$$F1score=\frac{2\times Precision\times Recall}{Precision+Recall}$$

This indicator allows one to fully evaluate the effectiveness of the algorithm. The optimal algorithm should have the largest possible value of the F1 metric. F1 ranges from 0 to 1. An F1 score of 1 indicates perfect precision and recall (best performance), and a score of 0 indicates that either precision or recall is 0 (worst performance).

The true positive rate (TP) shows the number of days with anomalies detected by the algorithm on the days when these anomalies actually occurred. Our data contain 3 days of anomalies identified by experts during data analysis. The exact time (hour, minute, and second) of occurrence of the anomaly is unknown. Therefore, TP can take values from 0 to 3. In addition, for each algorithm, a type II error was calculated—a false negative (FN) rate, because in this case, the algorithm does not detect an anomaly if it exists, which is critical for a biological early warning system. Type I error—FP (false positive) rate shows how many times our algorithm has flagged data points that are not actually true anomalies. It could take values from 0 to the amount of test data if the algorithm marked all data as anomalies on a test day when there was no anomaly.

The hyperparameter contamination percentage (contamination rate), i.e., the proportion of outliers in the sample is used in one form or another in all considered methods [40,41]. Data analysis was carried out in Python programming language (V3.9.12) using the scikit-learn machine learning package (V 1.0.2) [42].

## 4. Results

### 4.1. Elliptic Envelope

The elliptic envelope algorithm creates an imaginary elliptical region around the dataset. Data falling within this region is considered normal data, and anything outside the range is returned as outliers (anomalous). The algorithm works best if the data have Gaussian distribution. The elliptic envelope method uses the covariance estimate with Mahalanobis distance [40,43]. This model is implemented in the elliptic envelope function of the scikit-learn covariance module [42]. The function parameters we used for the simulation (excluding contamination rate): store_precision = true, assume_centered = false, support_fraction = none (the proportion of points to be included in the support of the raw MCD estimate).

In addition, the influence of feature standardization on model performance was studied [44]. Centering and scaling were performed independently for each feature by calculating the corresponding statistics from the samples in the training set. The obtained values of the mean value and standard deviation were then used on the test data. Standardized the features using the StandardScaler class from the preprocessing module of the scikit-learn library.

The F1 score equal to one was obtained when averaging the mollusk activity data for 15 min with a contamination rate of less than 0.0005 and for 5 min with a contamination rate in the range from 0.0005 to 0.001 (Figure 4). At the same time, the results of estimating the F1 score with standardized data and without standardization (not shown) for the elliptic envelope model are the same. Without averaging and a contamination rate less than 0.0001, as well as one-minute averaging and a contamination rate equal to 0.0005, the model showed a false negative result equal to two, i.e., the model does not detect an anomaly in two out of three cases (Figure 4). There are also parameters under which the model shows a single false negative result, for example, with five-minute averaging and a contamination rate less than or equal to 0.0001.

### 4.2. Isolation Forest (iForest)

One effective way to anomaly detection in multivariate datasets is to use the random forest algorithm. The isolation forest “isolates” observations by randomly choosing a feature and then randomly choosing a separation value between the maximum and minimum values of the selected feature [45]. The implementation of the isolation forest algorithm is based on an ensemble of extremely randomized tree regressors [42]. Data points are isolated by splitting the data multiple times until each data point is isolated. The path length of trees in the forest, which depends on the height of the tree and the average height of the forest, is taken as an anomaly score [46]. According to the results of Liu et al. [46], the maximum depth of each tree is set to ⌈log2(n)⌉, where n is the number of samples used to build the tree. To tune the iForest algorithm, a search was carried out using the following hyperparameters: the average value, the number of samples (n), and the number of trees (T), as well as with and without data normalization. The remaining parameters of the algorithm were as follows: max_features = 1.0 (the number of features to draw from X to train each base estimator), bootstrap = false (sampling without replacement is performed), warm_start = false.

Figure 5 shows the results of the iForest algorithm for n = 256 and data normalization. Data normalization was carried out using the scikit-learn MinMaxScaler library, which transforms each function individually so that it is in a given range on the training set, for example, between zero and one. The authors of the algorithm Liu et al. [46] showed that n = 256 and T = 100 are optimal for a wide range of tasks. However, for our dataset, the best result at n = 256 (F1 score = 1) was obtained with the number of trees T = 5, averaging over 30 min, and a contamination rate of less than 0.001 (Figure 5c). With a fixed number T = 5 and averaging the data over 15 min, the best result was obtained with n = 150 and a contamination rate less than 0.001 (Figure 5d). With T = 50, averaging the data over 15 min, the best result is obtained with n = 70 and a contamination rate of less than 0.001 (Figure 5e).

### 4.3. One-Class SVM

The SVM algorithm was introduced by Schölkopf et al. [47] and implemented in the “Support Vector Machines” module in the svm.OneClassSVM. To determine the boundary, the choice of a kernel and a scalar parameter is required. The RBF core is usually chosen, although there is no exact formula or algorithm for setting its throughput parameter. This is the default value in the scikit-learn library. The nu parameter, also known as the one-class SVM margin, corresponds to the probability of finding a new but regular observation outside the model boundary.

SVM is typically used for supervised learning, but one-class SVM can be used for anomaly detection in unsupervised learning. The goal of the SVM algorithm is to find the maximum margin hyperplane in an N-dimensional space (N is the number of features) that clearly classifies the data points [47,48]. In the case of using SVM for anomaly detection, the task is to find a function that is positive for areas with high point density and negative for areas with low point density. When setting up the one-class SVM model, we iterated over the values of the hyperparameter nu, which is the upper bound on the proportion of learning errors and the lower bound on the proportion of support vectors, changed the kernel type (rbf, sigmoid, or poly) and the coefficient of the kernel function γ, which affects the “smoothness” of the model. The rest of the model parameters were as follows: degree = 3 (degree of the polynomial kernel function), coef0 = 0.0 (independent term for poly and sigmoid kernels), tol = 0.001 (tolerance for stopping criterion), shrinking = True, cache_size = 200 (the size of the kernel cache) and max_iter = −1 (hard limit on iterations within solver).

The application of the one-class SVM algorithm for anomaly detection showed unsatisfactory results. The values of the F1 score when averaging the normalized and non-normalized data with the RBF kernel did not exceed 0.2 (Figure 6a,b). The best results of applying the one-class SVM algorithm were obtained using the sigmoid kernel function and the polynomial kernel function. F1 score values reach 0.55 with γ equal to 0.05, the poly kernel type with nu 0.001 and 0.005, and with the kernel type sigmoid and γ = 0.001 with nu 0.005 (Figure 6c).

### 4.4. Local Outlier Factor (LOF)

The LOF algorithm was first described by Breunig et al. [49]. The anomaly score of each sample is called the local outlier factor. It measures the local density deviation of a given sample relative to its neighbors. It is local in the sense that the anomaly score depends on how isolated the object is in relation to the environment. More precisely, locality is given by k nearest neighbors, the distance from which is used to estimate the local density. By comparing the local density of a sample with the local densities of its neighbors, it is possible to identify samples that have a significantly lower density than their neighbors (they are considered outliers). The algorithm hyperparameters that we used (excluding k nearest neighbors) were algorithm = ball_tree (algorithm used to compute the nearest neighbors), leaf_size = 30, metric = Minkowski (metric to use for distance computation), *p* = 2 (parameter for the Minkowski metric, the standard Euclidean distance).

F1 score values equal to one were achieved by averaging the data for 1 and 5 min. At the same time, the best results for assessing the F1 score were obtained without standardization or normalization of data (Figure 7).

The F1 score values equal to one were obtained with the number of neighbors 100 and averaging the data for 1 min. When the number of neighbors is 120 and the contamination rate is less than or equal to 0.0001, an error FN = 1 occurs, i.e., the model does not detect an anomaly, despite its presence (Figure 8).

## 5. Discussion

Traditionally, changes in behavioral parameters (e.g., mean amplitude and frequency of clamping) among pollutant-exposed and unexposed groups of bivalves have been studied using statistical tools such as analysis of variance [7,50,51]. The field of machine learning provides methodologies ideally suited to the task of extracting knowledge from complex and multivariate animal behavior datasets with non-linear dependencies and unknown interactions between multiple variables [25]. In addition to the obvious ways to improve data analysis or control of experimental conditions, machine learning will provide new insights into the functioning of biological systems and the process of how and why these functions evolved [52]. Our results show that natural and technical anomalies in bivalve activity datasets can be detected using machine learning algorithms. The use of algorithms such as elliptic envelope, iForest, and LOF with a certain set of hyperparameters (averaging, scaling, etc.) allows one to select an anomaly without false alarms, i.e., to obtain an F1 score equal to one. F1 score is the harmonic mean of precision and recall and gives a better measure of the incorrectly classified cases than the Accuracy Metric. We have imbalanced class distribution exists and thus F1-score is a better metric to evaluate our models on. For example, for the IForest algorithm (n_estimators = 50, max_samples = 256, and outliers_fraction = 0.0005) the score F1 will be 0.6, but accuracy will tend to 1. Unsatisfactory results (F1-score < 0.2) were obtained using the SVM algorithm. Confusion matrices for each model at different hyperparameter settings are shown in Appendix A. In addition, unlike most studies, where the results are based on laboratory data [53], our results were obtained from data from the places of potential installation of such systems directly in the environment.

The main hyperparameter of our models, the contamination parameter, controls the threshold for the decision function when a scored data point should be considered an outlier. It has no impact on the model itself. The model assigns all data points an outlier score, the n ∗ contamination rate points with the highest scores are then labeled as anomalies. As a result, if the contamination rate is set too high (e.g., >0.001 for iForest), it would force the model to misclassify points as anomalies. If it is set too low (e.g., <0.0001 for and elliptic envelope with some averages), the model might miss some anomalies and only take into account the most severe ones.

Since an F1 score equal to one was obtained for three algorithms with different hyperparameters, we compared the methods in response time and anomaly detection (Table 1). For three anomalies, the best speed of response to the anomaly was shown by the IForest machine learning algorithm when averaging data over 15 min, T = 50 and n equal to 70. The averaging time in our work is considered a hyperparameter. For all models, the same averaging time set was studied: no averaging, 1 min, 5 min, 15 min, and 30 min. Since models use completely different anomaly detection principles, it is not possible to say in advance which averaging to use. The results of the best algorithms are presented in Table 1 in the manuscript. It can be seen from the table that for each algorithm the optimal is its own (certain) averaging time (for example, LOF—5 min, IForest—15 min). This confirms the correctness of our chosen strategy. The results once again confirm the need for careful selection of model hyperparameters.

For example, Figure 9 shows a comparison of the best detection time for the second anomaly by three algorithms. Vertical red bars are timestamps where the algorithms have identified anomalies. The LOF algorithm with hyperparameters data averaging 5 min, cr = 0.001, and k = 50 is 45 min behind the best result obtained by the iForest method when adjusting the model with hyperparameters data averaging 15 min, T = 50 and n = 70. The best response time of the model and detection of the second anomaly using the elliptic envelope algorithm is almost 10 h behind the detection time by the iForest algorithm (Figure 9).

In support of the successful application of machine learning algorithms, for example, LOF has been used for anomaly detection in a real-time fish farm water quality monitoring model [30]. Using data from a long-term ecological experiment with *Dreissena* mussels in freshwater ponds [54], the authors of [12] evaluated supervised and unsupervised machine learning algorithms to detect anomalies in the data. The results showed that supervised models perform better than unsupervised models, and unsupervised models show more variable levels of performance, confirming the importance of choosing the model structure and hyperparameters of unsupervised models [12].

## 6. Conclusions

By using unsupervised machine learning algorithms, the possibility of anomaly detection in bivalve data was evaluated in the work. We tested four machine learning algorithms for anomaly detection procedure: elliptic envelope, isolation forest (iForest), one-class support vector machine (SVM), and local outlier factor (LOF). Adjusting the hyperparameters of the models of four algorithms, estimates of their performances were obtained and the response time of methods to anomaly detection was estimated. The iForest algorithm showed the best result in anomaly detection and speed of its detection (with certain hyperparameter settings). The elliptic envelope and one-class SVM algorithms also showed good performance, but their anomaly detection rate turned out to be lower than that of algorithm iForest.

Thus, the machine learning algorithms proposed and studied in the work can be used for anomaly detection in the experimental data of mollusk activity for inclusion in the software of biological early warning systems to receive an alarm in real time. The ability of the system to respond to emergency situations and prevent the large-scale spread of negative impacts is important for sustainable management, assessment, and forecasting of the state of water bodies. In our work, we have considered four standard unsupervised machine learning algorithms for anomaly detection. Since our task is to further integrate the anomaly detection technique based on mollusk activity data into an existing and real-time device for monitoring the state of the aquatic environment, it is necessary to avoid increasing the computational complexity of the algorithms, increasing the load on the equipment and increasing the response time. Using the methods indicated in the work, we solved all the tasks. However, other unsupervised machine learning algorithms, such as DBSCAN, autoencoders, and principal component analysis, among others [55,56], are also commonly used for this task by researchers. Our future research will focus on exploring the potential of these algorithms in resolving the problem of anomaly detection in experimental data on mollusk activity and identifying behavior patterns in the activity of bivalves using clustering methods of unsupervised machine learning algorithms. This will improve the process for anomaly detection in mollusk activity data, enabling its integration into the software of the automated complex for monitoring aquatic environments.

## Figures and Tables

**Figure 1 sensors-23-02687-f001:**
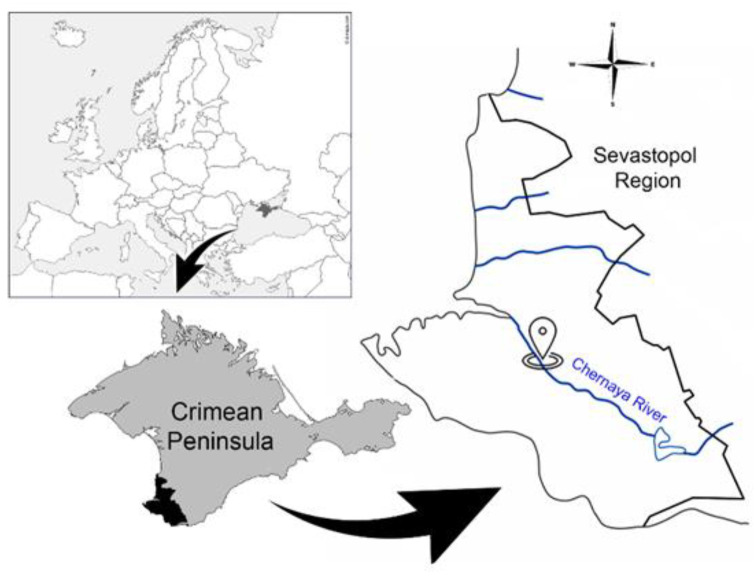
Location of the system for automated monitoring of the aquatic environment on Chernaya River (Sevastopol Region).

**Figure 2 sensors-23-02687-f002:**
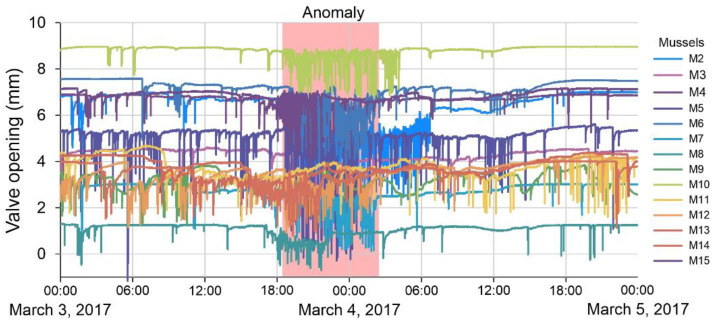
An example of the raw data on the activity of bivalves with anomaly (March 2017, Chernaya River).

**Figure 3 sensors-23-02687-f003:**
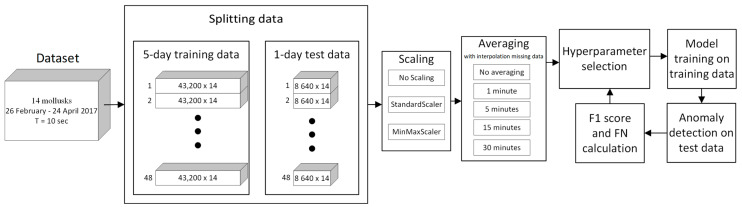
Blocks diagram of the algorithm.

**Figure 4 sensors-23-02687-f004:**
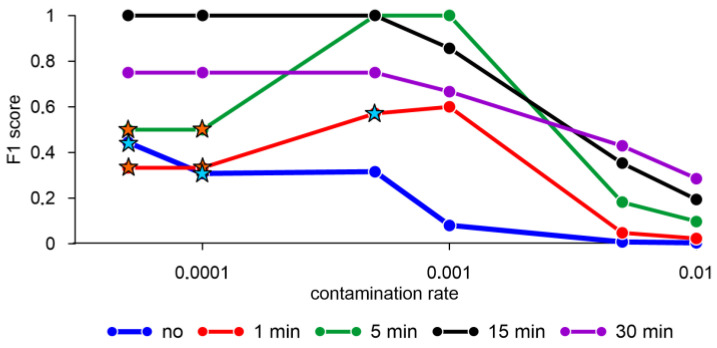
F1 score obtained by the elliptic envelope algorithm with different averaging of mollusk activity data and standardization of features. Orange stars—error FN = 1, blue stars—error FN = 2. Hereinafter the X scale is logarithmic.

**Figure 5 sensors-23-02687-f005:**
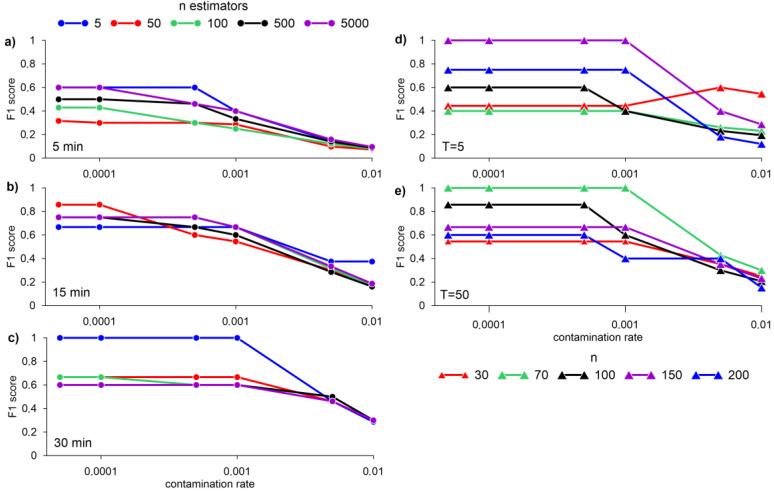
F1 score for different n-estimators with averaging data for 5 (**a**), 15 (**b**), 30 min (**c**), and normalizing using MinMaxScaler, max_samples = 256; different n, at T = 5 (**d**), and T = 50 (**e**) with 15-min data averaging obtained by the iForest algorithm.

**Figure 6 sensors-23-02687-f006:**
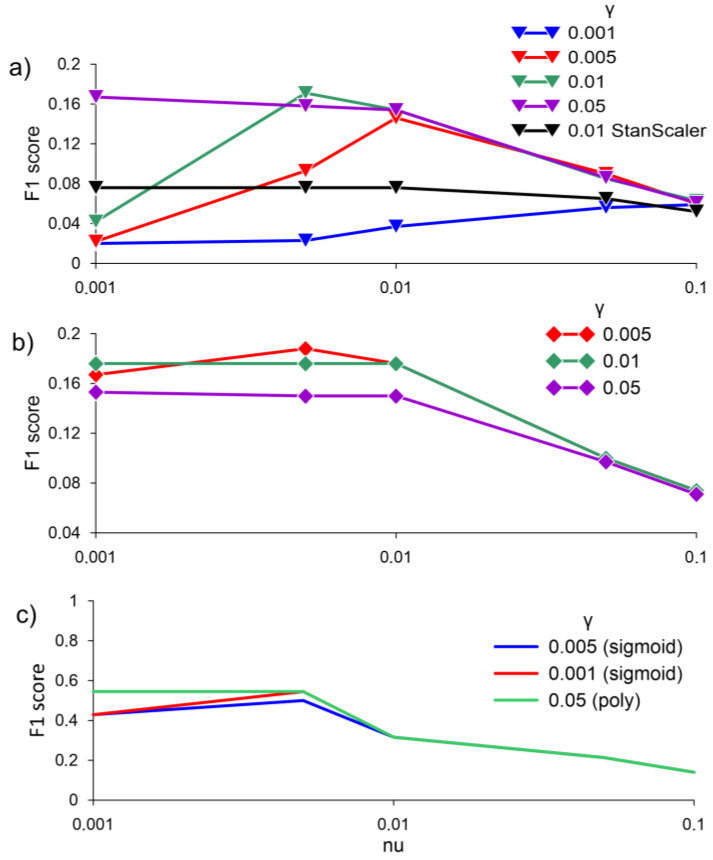
F1 score for (**a**) averaging data over 30 min, MinMaxScaler, and StandardScaler, with RBF; (**b**) averaging data over 30 min, without RBF scale; (**c**) with two kernels—sigmoid and poly, without scale at different values of γ, obtained by the one-class SVM algorithm.

**Figure 7 sensors-23-02687-f007:**
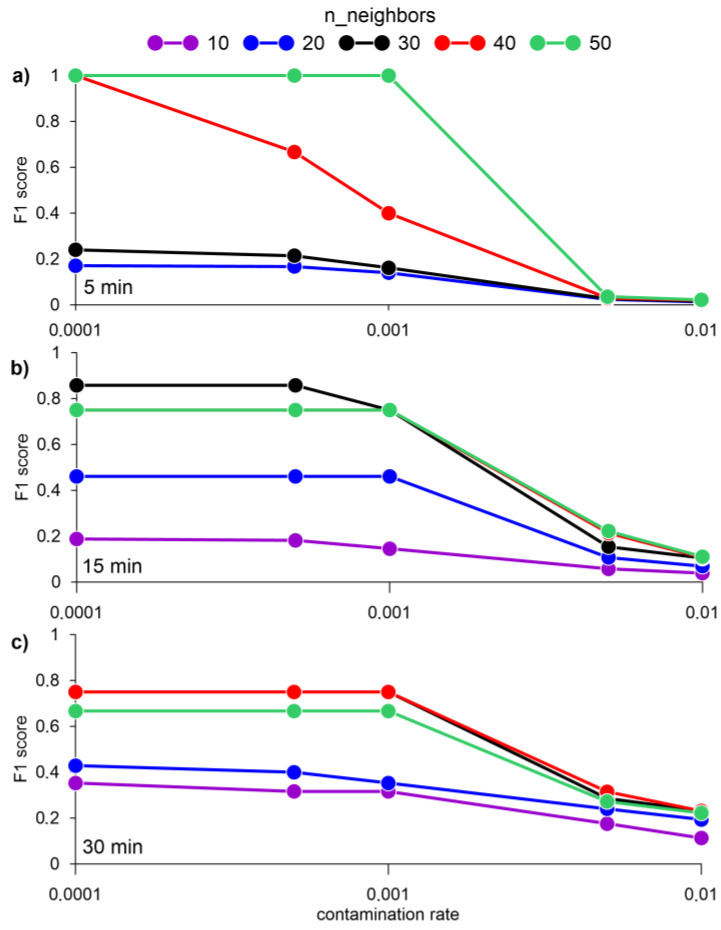
F1 score for a different number of nearest neighbors k and averaging data for 5 (**a**), 15 (**b**), and 30 min (**c**), obtained by the LOF algorithm.

**Figure 8 sensors-23-02687-f008:**
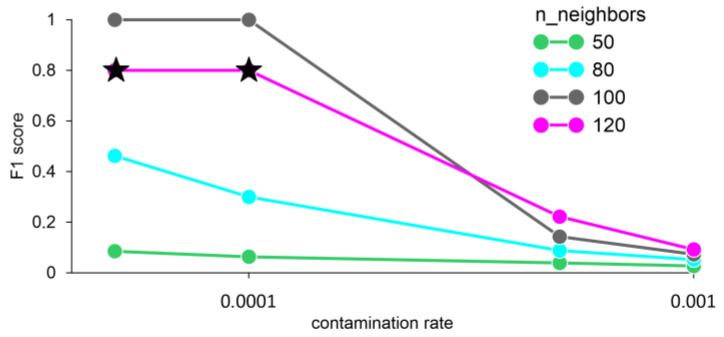
F1 score for different n_neighbors and averaging data for 1 min, obtained by the LOF method. Stars—FN error (false negative).

**Figure 9 sensors-23-02687-f009:**
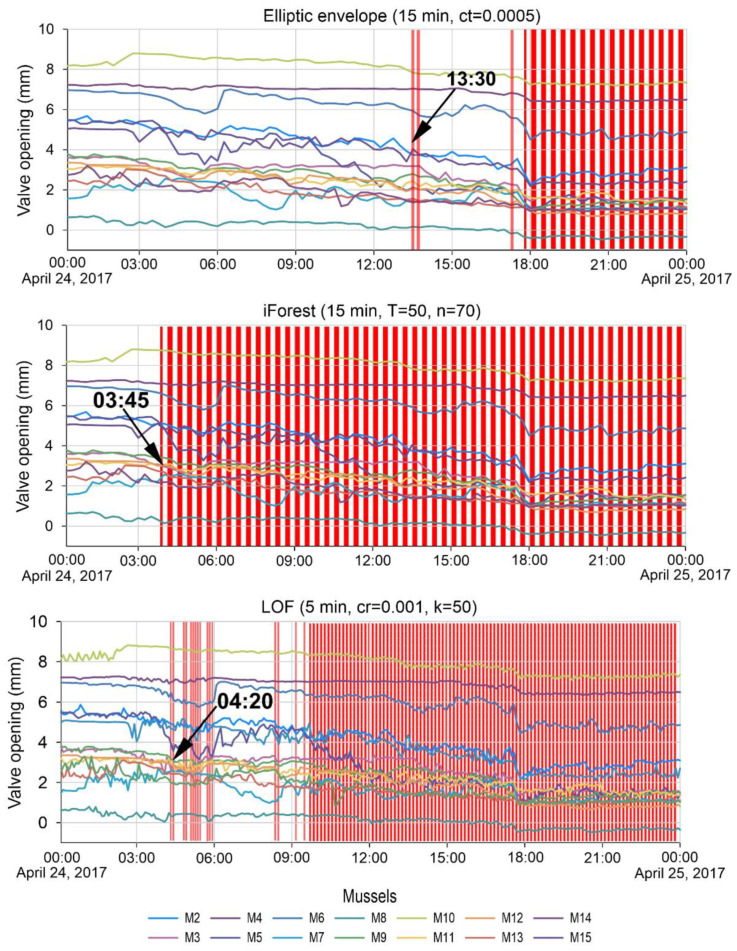
Detection time of the second anomaly by different algorithms.

**Table 1 sensors-23-02687-t001:** Comparison of algorithms by anomaly detection time.

Algorithm	Hyperparameters	Anomaly Detection Time		
		Anomaly 1	Anomaly 2	Anomaly 3
Elliptic envelope	Aver. 15 min, cr = 0.0005	17:45	13:30	18:30
	Aver. 15 min, cr = 0.00005	17:45	17:30	18:30
	Aver. 5 min, cr = 0.001	18:00	17:05	18:35
iForest	Aver. 30 min, T = 5, n = 256	19:00	04:00	18:30
	Aver. 15 min, T = 5, n = 150	18:15	03:45	18:45
	Aver. 15 min, T = 50, n = 70	17:15	03:45	18:15
LOF	Aver. 5 min, cr = 0.0001, k = 40	19:45	05:15	18:35
	Aver. 5 min, cr = 0.0001, k = 50	19:45	09:50	18:35
	Aver. 5 min, cr = 0.001, k = 50	19:25	04:20	18:35
	Aver. 1 min, cr = 0.0001, k = 100	19:51	05:12	18:34

Aver—averaging time, cr—contamination rate.

## Data Availability

Data available upon request.

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
