# Peer review of "Anomaly Detection in Biological Early Warning Systems Using Unsupervised Machine Learning"

_sensors, 2023, doi:10.3390/s23052687_

Round 1

Reviewer 1 Report

Dear Authors thanks for your good paper about Anomaly detection in biological early warning systems using unsupervised machine learning but it still need many corrections 

1- abstract is very poor writing

2- introduction is luck of information at the last paragraph please add the significant of your research 

3- methodology should make more clear about graphs and equations 

4- need to write conclusion in separate  part and make it in a good sentences because in your paper not clear what is the significant , background ...

Author Response

Response to Reviewer Comments

Dear Reviewer!

Thank you for your valuable comments! We tried to answer all your comments and clarify incomprehensible points. Below are the answers to your comments

Dear Authors thanks for your good paper about Anomaly detection in biological early warning systems using unsupervised machine learning but it still need many corrections 

Point 1: abstract is very poor writing

Response 1: abstract has been improved

Point 2: introduction is luck of information at the last paragraph please add the significant of your research 

Response 2: Information about the significance of our results has been added to the last paragraph in Introduction section

Point 3: methodology should make more clear about graphs and equations

Response 3: Information about the chart in the methodology section was added to the text of the article

Point 4: need to write conclusion in separate  part and make it in a good sentences because in your paper not clear what is the significant , background ...

Response 4: Сonclusions are highlighted in a separate section. Information about the significance of our results has been added to the text of the article

Reviewer 2 Report

This paper considers the possibility of using machine learning algorithms to anomaly detection in the behavioral reactions of mollusks in systems. In this paper, four unsupervised machine learning methods elliptic envelope, Isolation forest (iForest), One class support vector machine (SVM), and local outlier factor (LOF) are used to detect an emergency signal in the activity of bivalves.

However, there are some problems in this paper. The specific problems are as follows:

1.This paper mainly describes the existing methods, which cannot show the innovation of the paper.

2.The description of relevant work in the paper is vague, which cannot show the difference of their research.

3.Why this paper does not use the latest unsupervised machine learning methods for comparison?

4.This paper does not give other simulation parameters and future research contents.

5.Most importantly, the method content of this paper is more traditional, and does not reflect the theoretical depth and innovation.

Author Response

Response to Reviewer Comments

Dear Reviewer!

Thank you for your valuable comments! We tried to answer all your comments and clarify incomprehensible points. Below are the answers to your comments

This paper considers the possibility of using machine learning algorithms to anomaly detection in the behavioral reactions of mollusks in systems. In this paper, four unsupervised machine learning methods elliptic envelope, Isolation forest (iForest), One class support vector machine (SVM), and local outlier factor (LOF) are used to detect an emergency signal in the activity of bivalves.

However, there are some problems in this paper. The specific problems are as follows:

Point 1: This paper mainly describes the existing methods, which cannot show the innovation of the paper.

Response 1: The innovation of our work lies in the application of traditional algorithms to raw experimental data on mollusk activity in order to anomaly detection. We do not aim to develop new algorithms, but only to evaluate existing algorithms in solving this problem.

Point 2: The description of relevant work in the paper is vague, which cannot show the difference of their research.

Response 2: The description of relevant work was added to the Introduction

Point 3: Why this paper does not use the latest unsupervised machine learning methods for comparison?

Response 3: We do not aim to develop new algorithms, but only to evaluate existing algorithms in solving this problem. In our work, we have considered four standard unsupervised machine learning algorithms for anomaly detection. However, researchers use many other unsupervised ma-chine learning algorithms for this task, such as DBSCAN, Autoencoders, Principal Component Analysis, and others. Our further research will be devoted to the possibilities of using these algorithms to solve the problem of detecting anomalies in experimental data on mollusk activity.

Point 4: This paper does not give other simulation parameters and future research contents.

Response 4: We have added to the text of the article a description of other parameters used in the simulation and our future research contents.

Point 5: Most importantly, the method content of this paper is more traditional, and does not reflect the theoretical depth and innovation.

Response 5: The paper considers the possibility of using four traditional unsupervised machine learning algorithms to anomaly detection in the behavioral reactions of mollusks in systems for automated biomonitoring of the aquatic environment. We do not aim to develop new algorithms, but only to evaluate existing algorithms in solving this problem.

The innovation of our paper is to use raw bivalve activity data (shell opening width in mm) to detect anomalies and their detection rate. The bulk of scientific articles (given in the Introduction) use data from physicochemical sensors to control the aquatic environment.

Round 2

Reviewer 2 Report

Although the innovation of this paper is to use raw bivalve activity data (shell opening width in mm) to detect anomalies and their detection rate, we think improved algorithm is necessary to address these specific data. Otherwise, this paper will be rejected.

Author Response

Dear Reviewer,

The objective of our research is to detect anomalies in the raw behavioral data of bivalve mollusks using machine learning algorithms, with the aim of incorporating this technique into the software of an established monitoring system for aquatic environments. Through the utilization of the algorithms described in our work, we have successfully achieved this goal.

While it is true that the improvement of machine learning methods, such as using more complex algorithms, may increase the computational complexity and hardware requirements, it is not necessary in our case. Our aim is to integrate a reliable and efficient anomaly detection technique based on mollusk activity data into a real-time monitoring system, without introducing unnecessary complexities.

It is important to note that our approach is not limited to a single machine learning algorithm. Rather, we have developed a unique algorithm that incorporates various parameters, such as the number of days, in addition to machine learning techniques. Our approach is depicted in Figure 2, which illustrates that a specific machine learning method is just one component of our overall anomaly detection algorithm.
